# Global epidemiology of hepatitis C virus in dialysis patients: A systematic review and meta-analysis

Raoul Kenfack-Momo[1], Marceline Djuidje Ngounoue[1], Sebastien Kenmoe[2]*, Guy Roussel Takuissu[3], Jean Thierry Ebogo-Belobo[4], Cyprien Kengne-Ndé[5], Donatien Serge Mbaga[6], Elisabeth Zeuko'o Menkem[7], Robertine Lontuo Fogang[8], Serges Tchatchouang[9], Juliette Laure Ndzie Ondigui[6], Ginette Irma Kame-Ngasse[4], Josiane Kenfack-Zanguim[1], Jeannette Nina Magoudjou-Pekam[1], Arnol Bowo-Ngandji[6], Maimouna Mahamat[10,11], Seraphine Nkie Esemu[2], Lucy Ndip[2], Richard Njouom[12]

1 Department of Biochemistry, The University of Yaounde I, Yaounde, Cameroon, 2 Department of Microbiology and Parasitology, University of Buea, Buea, Cameroon, 3 Centre for Food, Food Security and Nutrition Research, Institute of Medical Research and Medicinal Plants Studies, Yaounde, Cameroon, 4 Medical Research Centre, Institute of Medical Research and Medicinal Plants Studies, Yaounde, Cameroon, 5 Epidemiological Surveillance, Evaluation and Research Unit, National AIDS Control Committee, Douala, Cameroon, 6 Department of Microbiology, The University of Yaounde I, Yaounde, Cameroon, 7 Department of Biomedical Sciences, University of Buea, Buea, Cameroon, 8 Department of Animal Biology, University of Dschang, Dschang, Cameroon, 9 Scientific Direction, Centre Pasteur du Cameroun, Yaoundé, Cameroon, 10 Faculty of Medicine and Biomedical Sciences, University of Yaoundé 1, Yaoundé, Cameroon, 11 Hemodialysis Unit, Yaoundé General Hospital, Yaoundé, Cameroon, 12 Virology Department, Centre Pasteur of Cameroon, Yaoundé, Cameroon

* ken_sebas@yahoo.fr

**Data Availability Statement:** All relevant data are within the manuscript and its Supporting Information files.

## Abstract

Dialysis is a replacement therapy for patients with End-Stage Renal Disease (ESRD). Patients on dialysis are at high risk of acquiring hepatitis C virus (HCV), which has become a leading cause of morbidity and mortality in this population. There is a wide range of prevalence of HCV in dialysis populations around the world. It is still unknown how prevalent HCV infection is among worldwide dialysis patients (including those undergoing hemodialysis and peritoneal dialysis). A review was conducted to estimate the global epidemiology of hepatitis C in dialysis patients. We searched PubMed, Excerpta Medica Database (Embase), Global Index Medicus and Web of Science until October 2022. A manual search of references from relevant articles was also conducted. Heterogeneity was evaluated by the $\chi^2$ test on Cochrane's Q statistic, and the sources of heterogeneity were investigated using subgroup analysis. In order to assess publication bias, funnel plots and Egger tests were conducted, and pooled HCV prevalence estimates were generated using a DerSimonian and Laird meta-analysis model. The study is registered with PROSPERO under CRD42022237789. We included 634 papers involving 392160 participants. The overall HCV case fatality rate was 38.7% (95% CI = 28.9–49). The global prevalence of HCV infection in dialysis population group were 24.3% [95% CI = 22.6–25.9]. As indicated by UNSD region, country, dialysis type, and HCV diagnostic targeted; Eastern Europe had the highest prevalence of 48.6% [95% CI = 35.2–62], Indonesia had 63.6% [95% CI = 42.9–82], hemodialysis patients had 25.5% [95% CI = 23.8–27.3], and anti-HCV were detected in 24.5%

**Funding:** This project is part of the EDCTP2 programme supported by the European Union under grant agreement TMA2019PF-2705. Grant Recipient: Dr Sebastien Kenmoe. The funders had no role in study design, data collection and analysis, decision to publish, or preparation of the manuscript.

**Competing interests:** The authors have declared that no competing interests exist.

[95% CI = 22.8–26.2]. Dialysis patients, particularly those on hemodialysis, have a high prevalence and case fatality rate of HCV infection. Hemodialysis units need to implement strict infection control measures.

## Introduction

Hepatitis C virus (HCV) is a small enveloped RNA virus, 55–65 nm in diameter, that targets the liver, causing mild inflammation (acute hepatitis) or severe and persistent inflammation (chronic hepatitis) [1]. HCV is the major cause of mortality from cirrhosis and/or hepatocellular carcinoma (HCC), and therefore a public health problem [2]. It is estimated that there are 58 million chronic HCV infected people worldwide, and 1.5 million new infections occur each year. As a blood-borne virus, HCV is primarily transmitted by blood. Depending on the geographical region, the transmission routes vary. Among these pathways, transfusions, intravenous drug use (IDU) and dialysis are most widely documented [2, 3]. As a result, dialysis would already be a route of HCV transmission, and needs to be emphasized more and more. Numerous studies have shown that dialysis patients are at high risk of acquiring HCV infections [4–6]. In a systematic review, hemodialysis (HD) significantly increases the likelihood of contracting HCV [7]. In another systematic review, high incidence rates of HCV infection were found in patients on HD [8]. Another study found that HD patients with HCV experience significantly worse quality of life (pruritus, depression, and anorexia) than HD patients without HCV [9]. Dialysis patients are increasingly at risk of HCV liver disease, which has become a major concern for health policy makers and care providers [10, 11]. In fact, individuals with End-Stage Renal Disease (ESRD) suffer from impaired immune systems, resulting in viral infections like HCV not clearing as effectively [12]. Increasing access to diagnosis and treatment is essential for reducing HCV-related mortality and morbidity in dialysis patients, according to WHO. The lack of epidemiological data that can map the pooled prevalence of HCV in worldwide dialysis centres and target those most in need of treatment will prevent this from being possible. HCV prevalence in different dialysis centres has been the subject of several systematic reviews. A number of these studies have been conducted in nations such as Afghanistan, India, Vietnam, China, Iran, Brazil, and Pakistan [7, 13–20].

Another has been conducted in the Middle East and North Africa, Latin America and Caribbean, Asia, and Africa regions [21–25]. A couple of narrative reviews (one on the worldwide magnitude of HCV in HD populations from 1999–2007; the other on HCV transmission in the dialysis setting and prevention strategies) are published [26, 27]. At the global level, two systematic reviews were published on HCV in HD patients. In the first one, the incidence rate (IR) was estimated between 1990–2012; while the most recent one examined the prevalence, associated risk factors, and mortality of hepatitis C infection in hemodialysis patients [8, 28]. Currently, no summary of evidence is available that reports the worldwide pooled prevalence of HCV in dialysis patients (including hemodialysis and peritoneal dialysis). Thus, the present systematic review and meta-analysis was conducted to determine the global prevalence of hepatitis C virus in dialysis patients.

## Materials and methods

### Data sources and search strategy

A guideline for reporting systematic reviews (S1 Checklist) guided the design of this review (S1 Table) [29]. The search strategy (S2 Table) was applied to four databases: Medline,

Excerpta Medica Database (Embase), Global Index Medicus, and Web of Science. The screening of all documents related to HCV infection among dialysis patients worldwide was restricted to English and French. Initially, we conducted our search in December 2021; however, we updated it in October 2022. In November 2022, we conducted manual searches of references from relevant articles and previous systematic reviews to identify any additional relevant references that may have been missed by online searches. A copy of the protocol was registered in PROSPERO under number CRD42022237789.

## Study selection

After applying our search strategy, articles were exported to Endnotes version X9 software, where duplicates were removed. RKM and SK independently selected articles based on their titles and/or abstracts on the Rayyan review platform. Fourteen reviewers obtained the full texts and screened potential eligible articles. Discussions and consensus were used to resolve disagreements between reviewers.

## Eligibility criteria

The included studies met the following criteria: 1) where cross-sectional, case control and cohort studies; 2) describe the diagnostic method used; 3) performed worldwide, and limited to human studies; 4) reporting HCV prevalence in hemodialysis and peritoneal units without regard to age, gender, sampling approach, ethnicity or clinical presentation, publication year, HCV detection method, or sample type; 5) reporting number of deaths in hemodialysis and peritoneal HCV positives patients. A study with fewer than 10 dialysis patients, case reports, and past reviews that did not provide data on HCV prevalence among dialysis patients were excluded. In addition, duplicates and studies without abstracts or full texts were excluded.

## Data extraction

A standardized Google form was used to extract the data independently. Among the extracted data were: the first name of the authors, the year of publication, the study design, the geographical location (UNSD region, WHO region, country income level), the sampling method, the study period, the inclusion criteria, the sample size, the type of dialysis (hemodialysis, peritoneal dialysis), the diagnostic technique used to detect HCV, the diagnostic target (HCV-Ab, HCV-Ag, HCV-RNA), the number of subjects infected by HCV, the number of HCV positive with known outcomes, the number of death in HCV positives patients, and participant characteristics (mean age, SD).

## Quality assessment

Using the method developed by Hoy et al. [30], we assessed the quality of the studies included. In this scale, two domains were assessed: internal (target population representation, sampling representation, random selection method, and data source) and external (appropriate inclusion criteria, size adequacy, reliability, validity, mode of sample collection, length of study period, reporting numerator(s) and denominator(s) for HCV prevalence) validity for a total score of 10 points. Study scores of 7–10, 3–6, and 0–3 points were classified as low, moderate, and high risk of bias, respectively (S3 Table).

## Data synthesis and analysis

A DerSimonian and Laird random-effects meta-analysis model with inverse variance method was used to pool study-specific estimates [31]. Cochrane's Q statistic is quantified by I2 values,

with I2 values of 25%, 50%, and 75% representing low, moderate, and high heterogeneity, respectively [32]. A funnel plot and Egger test were used to assess publication bias [33]. Sub-group analyses were performed to estimate prevalence variations by HCV diagnostic technique and target, type of dialysis, WHO region, UNSD region, country income level, country, study design, and sampling method. R software version 4.0.3 was used to conduct the analyses [34, 35], and a p value of 0.05 indicated a significant difference.

## Results

### Results of literature search

In the initial search, we identified 8441 articles, and 3044 duplicates were excluded (Fig 1). We obtained 177 articles from other sources. Based on the titles and abstracts, 4379 articles were excluded. The full-text screening of the remaining 1195 articles resulted in the exclusion of 561 studies for multiple reasons (S4 Table). In the end, 634 articles met our inclusion criteria and were included in the qualitative and quantitative synthesis (S1 Text).

### Characteristics of included studies

The characteristics of included studies are shown in S5 and S6 Tables. The majority of the 634 articles included in this review were cross-sectional studies (89.8%). Articles were published between 1990–2022, and participants were recruited between 1968–2022. Study participants ranged in mean age from 7.7 to 72.7 years, and most were recruited from hemodialysis centers (91.8%) and urban areas (57.9%). The male proportion ranged from 12 to 95.4%, the duration of dialysis was between 0.2 and 25.7 years, and the number of dialysis sessions per month was between 3 and 16. According to the WHO Region, most studies were conducted in Europe (34.4%), followed by Eastern Mediterranean (25.1%), Western Pacific (14.4%), America (12.6%), South-East Asia (9.8%) and Africa (2.7%). Most of the studies were conducted in high-income countries (48.7%), upper-middle-income countries (30.6%), lower-middle-income countries (17.5%), and low-income countries (2.4%). Most studies were conducted in Italy (11%), India (7.3%), Iran (7.1%), Japan (6.3%), Brazil (5.8%), and Saudi Arabia (5.4%). The most common HCV diagnostic method was indirect ELISA (66.6%), and most studies indicated a moderate risk of bias (65.9%) (S7 Table).

### Case fatality rate estimate of hepatitis C virus infections in dialysis peoples

According to Fig 2, an overall HCV fatality rate of 38.7% (95% CI; 28.9–49) was obtained from 490 HCV-positive dialysis participants recruited from eight studies conducted in eight countries (Morocco, Saudi Arabia, India, Italy, Japan, America, Iran, and China). A high level of heterogeneity was observed between studies that estimated HCV fatality rates (I2 = 77.5% [55.6%; 88.7%], p <0.001). Egger's regression test result (Table 1) and funnel plot (S1 Fig) showed no publication bias (P = 0.221). Studies with a low bias risk and cross-sectional designs produced similar results in sensitivity analyses.

### Global prevalence estimates of hepatitis C virus in dialysis patients

Based on a pooled analysis of 392160 dialysis patients, the prevalence of HCV infection was estimated at 24.3% [95% CI = 22.6–25.9], with considerable heterogeneity (I2 = 99.3% [99.2–99.3], p < 0.001) (Fig 3, Table 1, S2 Fig). The Egger's test and funnel plot indicated statistically significant publication bias (P = 0.001) (S3 Fig). Anti-HCV prevalence was 24.5% [95% CI = 22.8–26.2]; HCV core antigen (HCV-core Ag) was 16.7% [95% CI = 3.4–37]; anti-HCV/HCV-core Ag was 22.5% [95% CI = 14.6–31.6]; and viral RNA was 23% [95% CI = 17.9–28.7].

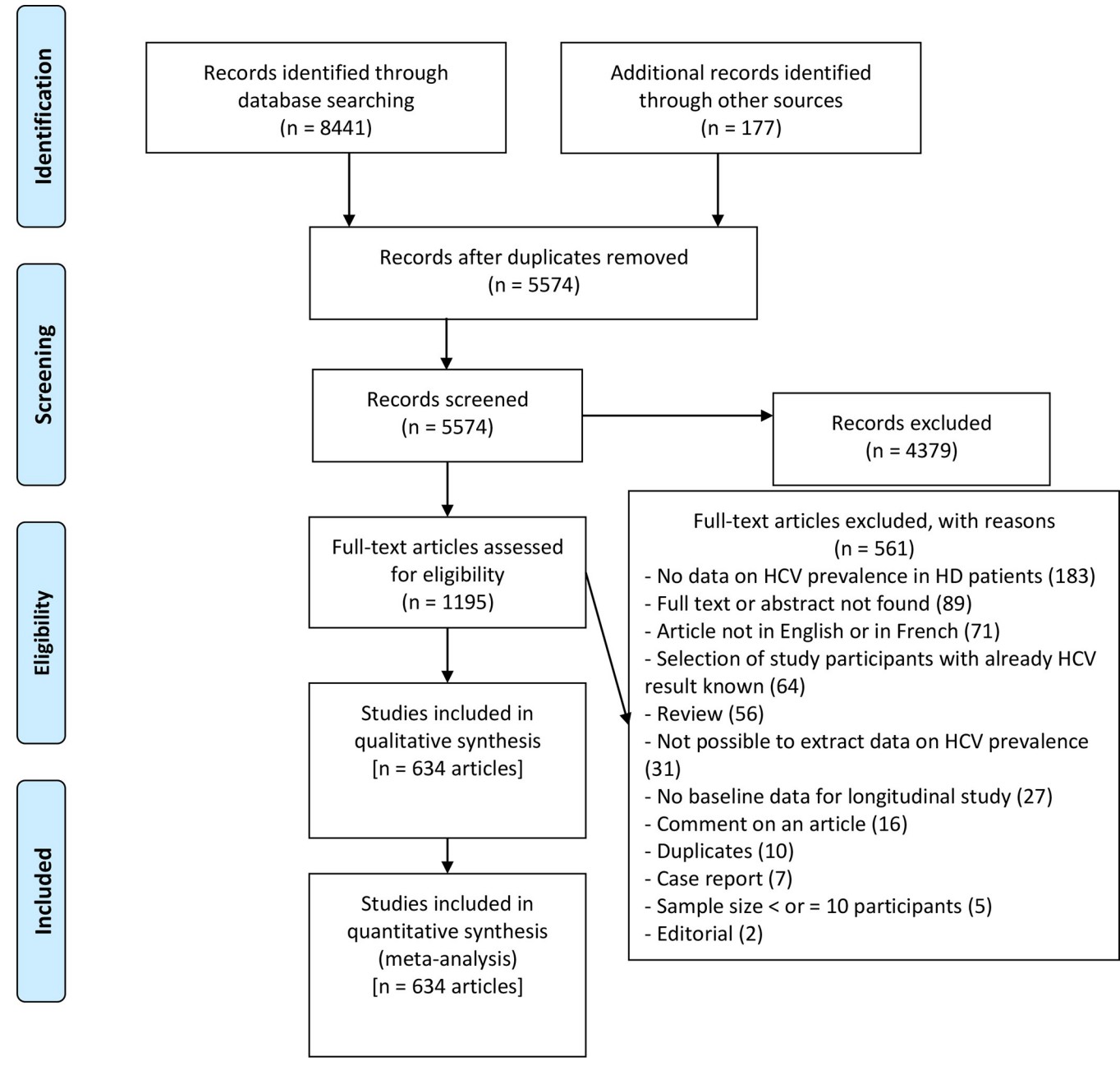

**Fig 1. Flow diagram for retrieving and processing article selection.**

## Subgroup analyses

A subgroup analysis was conducted to estimate prevalence variations based on HCV diagnostic technique and target, type of dialysis, WHO region, UNSD region, income level, country, study design, and sampling approach (S8 Table). Indonesia (63.6%, 95% CI = 42.9–82), Romania (55.3%, 95% CI = 30.8–78.5), Egypt (55.2%, 95% CI = 47.1–63.2), Kosovo (48.2%, 95% CI = 41–55.5), and Poland (46.7%, 95% CI = 37.9–55.5) had the highest prevalence rates (p<0.001) (Fig 4). Based on WHO region, prevalence was significantly higher in Eastern Mediterranean (27.9%; 95% CI: 24.3–31.6), South-East Asia (25.3%; 95% CI: 16.2–35.6) and Europe

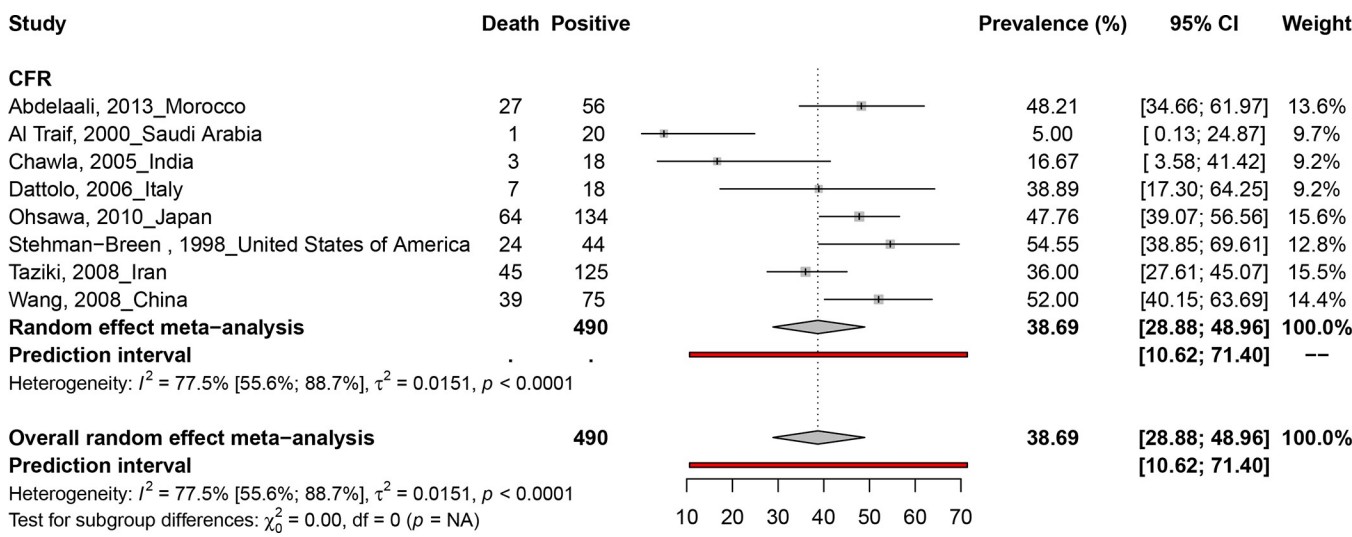

**Fig 2. Case fatality race of hepatitis C virus infections in dialysis population.**

(24.6%; 95% CI: 21.9–27.4) (p<0.001). Among UNSD Regions, Eastern Europe (48.6%; 95% CI = 35.2–62), Southeastern Asia (35.5%; 95% CI = 16.5–57.2), Western Asia (34.9%; 95% CI = 29.3–40.6), and Northern Africa (32.9%; 95% CI = 25.7–40.5) had the highest prevalence. Other significant differences were observed based on the type of dialysis (higher prevalence in hemodialysis, p = 0.001) and HCV diagnostic method (higher prevalence when using indirect ELISA, p = 0.003).

## Discussion

This systematic review and meta-analysis aimed to determine the global epidemiology of hepatitis C virus in dialysis patients. In total, 634 studies were included, published between 1990 and 2022, and conducted on 392160 dialysis patients. A case fatality rate of 38.7% was obtained from eight studies. The global prevalence of HCV infection in dialysis patients was 24.3%. In the Eastern Mediterranean, South-East Asia, and Europe WHO regions, as well as in Eastern Europe, Southeastern Asia, Western Asia, and Northern Africa UNSD regions, the prevalence was significantly higher.

**Table 1. Summary of meta-analysis results for epidemiology of hepatitis C virus in dialysis patients.**

| | Prevalence. % (95%CI) | 95% Prediction interval | N Studies | N Participants | ¶H (95%CI) | §$I^2$ (95%CI) | P heterogeneity |
|---|---|---|---|---|---|---|---|
| **HCV case fatality rate in HD** | | | | | | | |
| Overall | 38.7 [28.9–49] | [10.6–71.4] | 8 | 490 | 2.1 [1.5–3] | 77.5 [55.6–88.7] | <0.001 |
| Cross-sectional | 32.3 [18.4–47.9] | [0–86] | 5 | 294 | 2.5 [1.7–3.8] | 84.1 [64.2–92.9] | <0.001 |
| Low risk of bias | 46 [39.8–52.3] | [30.7–61.7] | 6 | 452 | 1.3 [1–2] | 37.7 [0–75.2] | 0.155 |
| **HCV prevalence in HD** | | | | | | | |
| Overall | 24.3 [22.6–25.9] | [0–70.8] | 667 | 392160 | 11.6 [11.5–11.8] | 99.3 [99.2–99.3] | <0.001 |
| Cross-sectional | 24.2 [22.4–26] | [0–72.7] | 597 | 317425 | 11.5 [11.3–11.6] | 99.2 [99.2–99.3] | <0.001 |
| Low risk of bias | 23.9 [21–26.8] | [0–72.6] | 229 | 246733 | 15.9 [15.7–16.2] | 99.6 [99.6–99.6] | <0.001 |

CI: confidence interval; N: Number; 95% CI: 95% Confidence Interval; NA: not applicable. ¶H is a measure of the extent of heterogeneity, a value of H = 1 indicates homogeneity of effects and a value of H >1indicates a potential heterogeneity of effects. §: I2 describes the proportion of total variation in study estimates that is due to heterogeneity, a value > 50% indicates presence of heterogeneity.

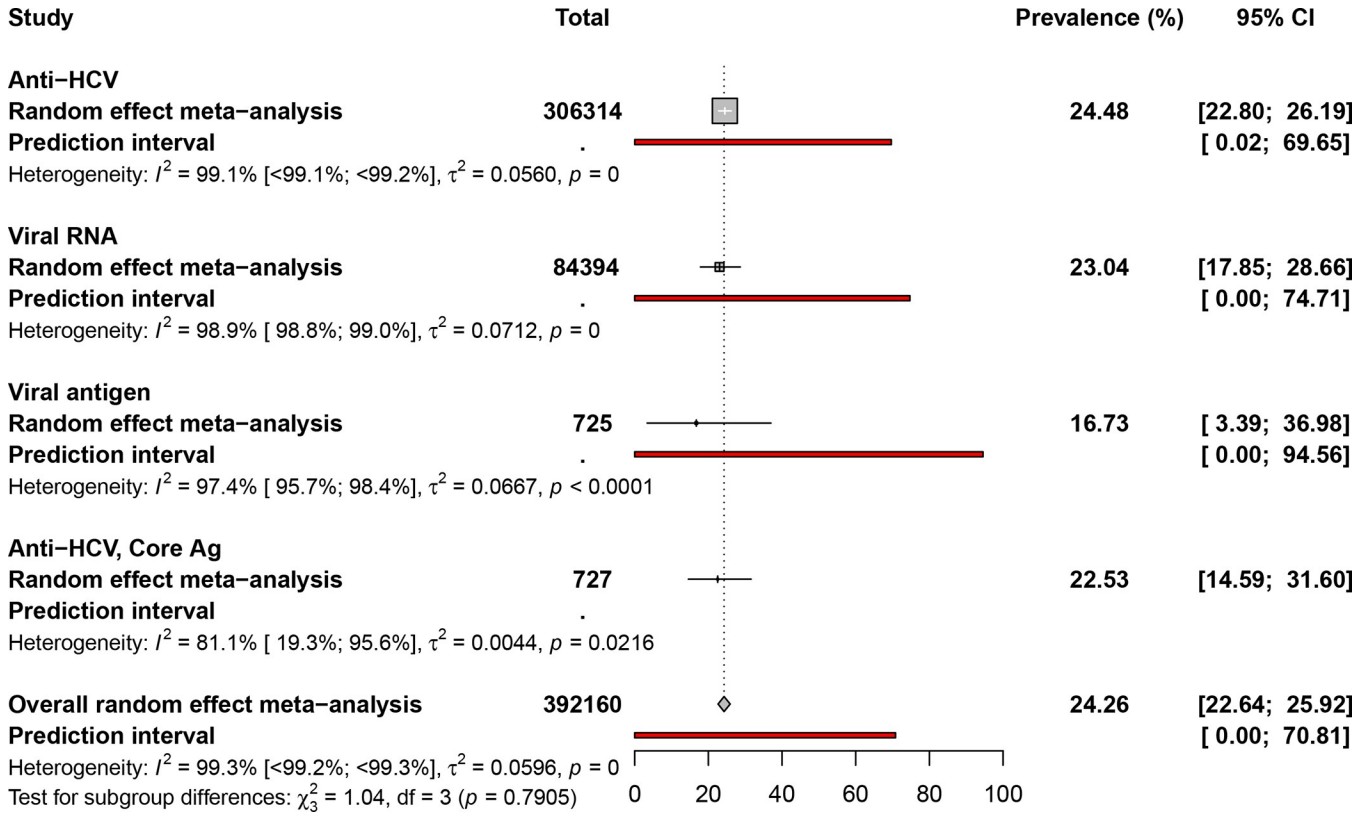

**Fig 3. Prevalence estimates of hepatitis C virus infection in dialysis patients.**

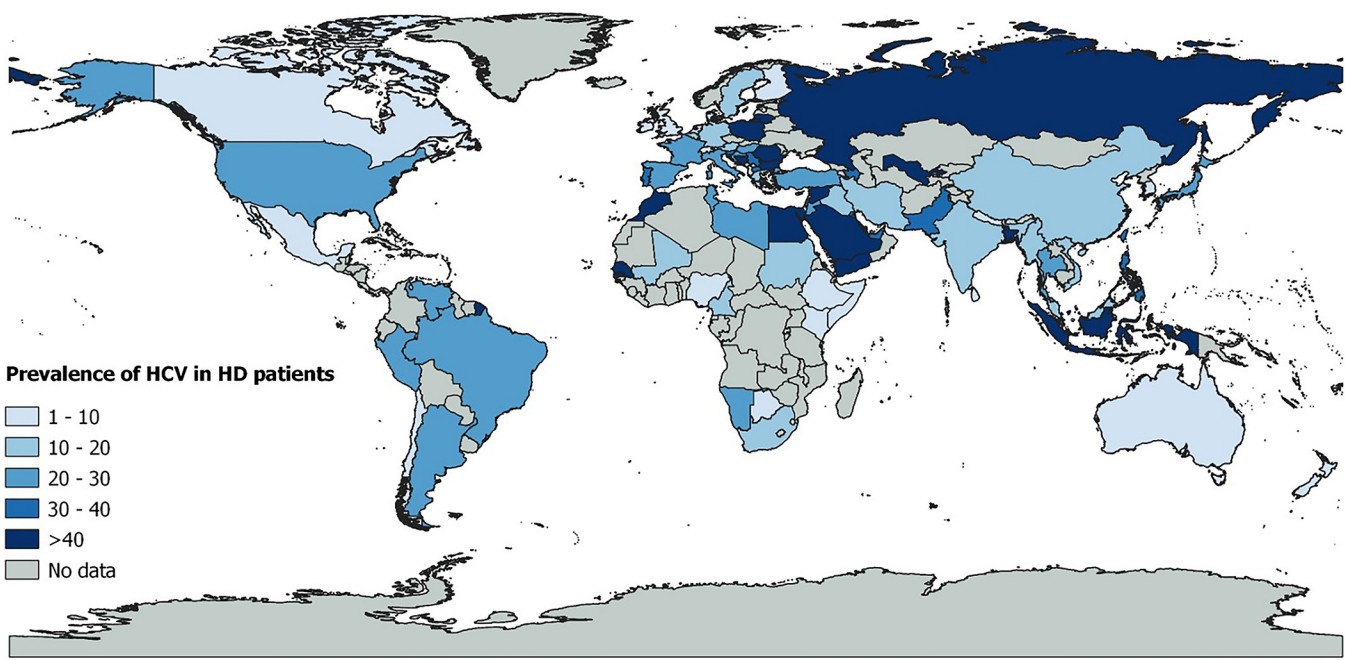

**Fig 4. Global prevalence of hepatitis C virus infection in dialysis patients.** The base map was taken from (https://www.naturalearthdata.com/) and modified with Qgis software.

The high HCV fatality rate estimated at 38.7% in this study is not surprising given that many studies have investigated the relationship between anti-HCV antibody and/or HCV-RNA positive status and survival in patients on long-term dialysis have found that mortality is very high in this patient group [36–39]. This excess death is not well understood, and there is active research underway to determine the reasons. The natural course of HCV in dialysis patients has not yet been investigated, even though it differs from other HCV positive patient groups [40]. In HCV-positive dialysis patients, worse quality of life (anemic complications and high hospitalizations) may explain increased mortality [9, 37]. It has also been suggested that HCV infection can adversely affect hemodialysis survival through direct effects such as liver disease (cirrhosis and hepatocellular carcinoma) [36, 41]. However, these direct effects remain unclear due to the lower survival rate of dialysis patients with HCV-positive cirrhosis and hepatocellular carcinoma (HCC) [42]. HCV infection may also negatively impact survival through indirect effects such as cardiovascular disorders, immune deficiencies, and sepsis [43, 44]. The number of articles that allowed us to determine HCV case fatality rate in this study was relatively small (08 articles) and the number of patients who were considered for this purpose was relatively small (N = 490 patients). Our recommendation is to focus on this goal in prospective studies.

The pooled HCV infection rate in world dialysis patients (24.3%) is very close to the 21% found by Greeviroj et al. in their recent meta-analysis [28]. Furthermore, it is fourteen times higher than the prevalence of hepatitis C in the general population, estimated at 1.8% in a recent systematic review [45]. Also, it is much higher than the global prevalence of HCV infection (0.87% in blood donors, 3.4% in men who have sex with men, 17.7% in prisoners, and 2·4% in people living with HIV) [26, 39, 46–50].

Among the settings was HCV is most commonly transmitted, dialysis centres were the most common. Several risk factors were suggested to explain this high prevalence of HCV in dialysis centres: immune response failure, high blood transfusions, inadequate screening of HCV for donated blood, no use of erythropoietin, contamination in the operational process of the dialysis environ, time on dialysis, high prevalence of HCV in dialysis unit, cross-contamination, contaminated hands of staff, understaffing, lack of medically qualified and competent personnel, repeated needle puncture, incomplete disinfection machine [14, 51–70]. Researchers have demonstrated that implementing preventive measures to control infection and a better transfusion policy on dialysis units could lead to declines in HCV prevalence among HD patients [71]. According to some studies, treating HCV-infected patients with dedicated machines could reduce HCV transmission in dialysis units [66, 72–77]. For other studies, isolation of HCV infected subjects in HD units is not required if universal infection control precautions are applied strictly [78–80]. Phylogenetic studies have suggested some mechanisms of HCV spread among hemodialysis patients. There is a high probability of nosocomial dissemination of HCV in dialysis units through cross-contamination (patient-to-patient transmission) [81–88]. Napoli et al. suggested diagnosing the infection during the acute phase to reduce the burden of HCV in hemodialysis patients [89]. Most studies (N = 561) included in this summary used anti-HCV antibodies as HCV diagnostics, which cannot distinguish between resolved and current infections. In addition, because of the long serological window, detecting HCV infection by targeting anti-HCV antibodies could be delayed, and viraemia could occur without HCV antibodies being detected [90–94]. Nucleic Acid Amplification Technology (NAT) is recommended by Kidney Disease Improving Global Outcomes (KDIGO) to diagnose HCV in dialysis patients [10]. NAT technology targeting at viral-RNA has several challenges (requires complex instrumentation), especially in resource-limited settings [95].

For several authors, the HCV-core Ag is more sensitive and specific than NAT techniques and may be an alternative to them [96–100]. But very few studies that determine HCV core

antigen prevalence in dialysis population have been carried out (only 04 in this report). The next studies should target HCV-core Ag to determine the precise prevalence of HCV and to better monitor the spread of HCV in dialysis units. We found that prevalence of anti-HCV antibodies, anti-HCV/HCV-core Ag and viral RNA are very close (24.5%, 22.5% and 23% respectively). Anti-HCV/HCV-core Ag is enzyme immunoassay enables to detect the presence of anti-HCV antibodies and/or Core viral antigen in sample. However, only 02 studies have allowed us to determine the prevalence of HCV in dialysis patients, using a 4th generation enzyme immunoassay (Monolisa HCV Ag-Ab), which may been overestimate. Several reports have shown that, most patients with anti-HCV antibodies in HD units have viral genomes [101–105]. Thus, Elzouki al believes that anti-HCV antibodies positive HD patients can be considered potentially infective [106]. Moreover, George et al. failed to confirm the presence of HCV viraemia in anti-HCV antibodies negatives HD patients, and the authors concluded that routine HCV RNA detection in anti-HCV-negative HD patients appears rather unnecessary [104]. However, in other reports, a poor correlation between anti-HCV antibodies and the presence of HCV-RNA was found [12, 106]. In this systematic review we did not performed analysis to determine the correlation between the two markers. Next summary should focus on it.

There are some limitations in the current study. First, our HCV pooled prevalence showed significant heterogeneity and publication bias in the current study. The case fatality rate estimates also showed heterogeneity. Secondly, in some regions and countries, the HCV prevalence in dialysis patients was determine from a single article, and this cannot be representative of the exact burden in these regions or countries.

Third, we only selected articles in English and French. We provided the first global pooled prevalence of HCV infection in dialysis patients (including hemodialysis and peritoneal dialysis) and estimate the case fatality rate of this virus in this population.

There is a high prevalence of HCV infection as well as a high case fatality rate in this study. With an increase in the number of dialysis patients requiring admission, strategies aimed at preventing and eradicating HCV in dialysis units are urgently needed.

## Supporting information

**S1 Checklist. Preferred reporting items for systematic reviews and meta-analyses checklist.**
(PDF)

**S1 Table. Preferred reporting items for systematic reviews and meta-analyses checklist.**
(PDF)

**S2 Table. Search strategy in databases.**
(PDF)

**S3 Table. Items for risk of bias assessment.**
(PDF)

**S4 Table. Main reasons of exclusion of eligible studies.**
(PDF)

**S5 Table. Characteristics of included studies.**
(PDF)

**S6 Table. Individual characteristics of included studies.**
(PDF)

**S7 Table. Risk of bias assessment.**
(PDF)

**S8 Table. Subgroup analyses of prevalence of hepatitis C virus in dialysis patients.**
(PDF)

**S1 Text. List of included studies.**
(PDF)

**S1 Fig. Funnel chart for publications of the hepatitis C virus case fatality rate in dialysis patients.**
(PDF)

**S2 Fig. Global prevalence estimate of hepatitis C virus infection in dialysis patients.**
(PDF)

**S3 Fig. Funnel chart for publications of the hepatitis C virus prevalence in dialysis patients.**
(PDF)

## Author Contributions

**Conceptualization:** Raoul Kenfack-Momo, Marceline Djuidje Ngounoue, Sebastien Kenmoe, Richard Njouom.

**Data curation:** Raoul Kenfack-Momo, Sebastien Kenmoe, Guy Roussel Takuissu, Jean Thierry Ebogo-Belobo, Cyprien Kengne-Ndé, Donatien Serge Mbaga, Elisabeth Zeuko'o Menkem, Robertine Lontuo Fogang, Serges Tchatchouang, Juliette Laure Ndzie Ondigui, Ginette Irma Kame-Ngasse, Josiane Kenfack-Zanguim, Jeannette Nina Magoudjou-Pekam, Arnol Bowo-Ngandji.

**Formal analysis:** Sebastien Kenmoe, Cyprien Kengne-Ndé.

**Funding acquisition:** Sebastien Kenmoe.

**Methodology:** Raoul Kenfack-Momo, Marceline Djuidje Ngounoue, Sebastien Kenmoe, Guy Roussel Takuissu, Jean Thierry Ebogo-Belobo, Cyprien Kengne-Ndé, Donatien Serge Mbaga, Elisabeth Zeuko'o Menkem, Robertine Lontuo Fogang, Serges Tchatchouang, Juliette Laure Ndzie Ondigui, Ginette Irma Kame-Ngasse, Josiane Kenfack-Zanguim, Jeannette Nina Magoudjou-Pekam, Arnol Bowo-Ngandji, Maimouna Mahamat, Seraphine Nkie Esemu, Lucy Ndip, Richard Njouom.

**Project administration:** Raoul Kenfack-Momo, Marceline Djuidje Ngounoue, Sebastien Kenmoe, Richard Njouom.

**Supervision:** Richard Njouom.

**Validation:** Raoul Kenfack-Momo, Marceline Djuidje Ngounoue, Sebastien Kenmoe, Guy Roussel Takuissu, Jean Thierry Ebogo-Belobo, Cyprien Kengne-Ndé, Donatien Serge Mbaga, Elisabeth Zeuko'o Menkem, Robertine Lontuo Fogang, Serges Tchatchouang, Juliette Laure Ndzie Ondigui, Ginette Irma Kame-Ngasse, Josiane Kenfack-Zanguim, Jeannette Nina Magoudjou-Pekam, Arnol Bowo-Ngandji, Maimouna Mahamat, Seraphine Nkie Esemu, Lucy Ndip, Richard Njouom.

**Writing – original draft:** Raoul Kenfack-Momo, Sebastien Kenmoe.

**Writing – review & editing:** Raoul Kenfack-Momo, Marceline Djuidje Ngounoue, Sebastien Kenmoe, Guy Roussel Takuissu, Jean Thierry Ebogo-Belobo, Cyprien Kengne-Ndé,

Donatien Serge Mbaga, Elisabeth Zeuko'o Menkem, Robertine Lontuo Fogang, Serges Tchatchouang, Juliette Laure Ndzie Ondigui, Ginette Irma Kame-Ngasse, Josiane Kenfack-Zanguim, Jeannette Nina Magoudjou-Pekam, Arnol Bowo-Ngandji, Maimouna Mahamat, Seraphine Nkie Esemu, Lucy Ndip, Richard Njouom.

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
