## [Decision Letter · Decision Letter 0]

23 Feb 2023

PONE-D-23-01990Global Epidemiology of Hepatitis C Virus in Dialysis patients: a systematic review and meta-analysis.PLOS ONE

Dear Dr. Kenmoe,

Thank you for submitting your manuscript to PLOS ONE. After careful consideration, we feel that it has merit but does not fully meet PLOS ONE’s publication criteria as it currently stands. Therefore, we invite you to submit a revised version of the manuscript that addresses the points raised during the review process.

We look forward to receiving your revised manuscript.

Kind regards,

Chen-Hua Liu

Academic Editor

PLOS ONE

2. We note that Figure 4 in your submission contain [map/satellite] images which may be copyrighted. All PLOS content is published under the Creative Commons Attribution License (CC BY 4.0), which means that the manuscript, images, and Supporting Information files will be freely available online, and any third party is permitted to access, download, copy, distribute, and use these materials in any way, even commercially, with proper attribution. For these reasons, we cannot publish previously copyrighted maps or satellite images created using proprietary data, such as Google software (Google Maps, Street View, and Earth). For more information, see our copyright guidelines: http://journals.plos.org/plosone/s/licenses-and-copyright.

a. You may seek permission from the original copyright holder of Figure 4  to publish the content specifically under the CC BY 4.0 license.   

Reviewers' comments:

Reviewer's Responses to Questions

**Comments to the Author**

1. Is the manuscript technically sound, and do the data support the conclusions?

Reviewer #1: Yes

Reviewer #2: No

Reviewer #3: Partly

2. Has the statistical analysis been performed appropriately and rigorously? 

Reviewer #1: Yes

Reviewer #2: No

Reviewer #3: I Don't Know

3. Have the authors made all data underlying the findings in their manuscript fully available?

Reviewer #1: Yes

Reviewer #2: Yes

Reviewer #3: No

4. Is the manuscript presented in an intelligible fashion and written in standard English?

Reviewer #1: Yes

Reviewer #2: No

Reviewer #3: Yes

5. Review Comments to the Author

Reviewer #1: Thanks for this comprehensive review of the global epidemiology of hepatitis C virus in dialysis patients. There is no major issue to modify.

[minor problem]

1. The abbreviation could be shown in the first place (hepatocellular carcinoma [HCC], line 12, page 17)

Reviewer #2: The authors reported a systemic review and meta-analysis on global epidemiology of hepatitis C virus in dialysis patients. They included 634 papers and concluded that hemodialysis cases have a high prevalence and case fatality rate. They also showed the significant regional variation, and the included papers had a moderate risk of bias, and high index of heterogeneity. The authors targeted an important issue and the results had potentially valuable information for the regional public health issue. However, several issues need to be addressed before further consideration.

Major issue:

1. The high case fatality rate is in line with the well-known high mortality and poor outcome of the HCV dialysis patient. However, little information was provided in the method part. For example, only 8 papers of the 634 papers were included in this part of analysis. More information and discussion about the paper selection criteria, research type, data extraction, data of hospital mortality or long-term mortality need to be addressed.

2. Numerical errors and inconsistence need a thorough review of the whole article to verify data and made correction. For example, third line of the Discussion part: “392.160” , line 111 “eight studies conducted in seven countries”, in the result table: “8” study, in the discussion ”The number of articles that allowed us to determine HCV case fatality rate in this study was relatively small (07 articles).

3. About 75% of anti-HCV dialysis patient were viremia, and the HCV core antigen method is a relative new methodology. The presented data showed similar prevalence of Anti-HCV, HCV core antigen and HCV RNA without further analysis and discussion may lead to misunderstanding for the readers.

Reviewer #3: The authors did a meta-analysis on the global epidemiology of hepatitis C virus (HCV) infection in dialysis patients, they found that the global prevalence of HCV infection in the dialysis population was 24.3%, the overall case fatality rate was 38.7%, while Eastern Europe and Indonesia had the highest prevalence rates among regions and countries respectively. They concluded that dialysis particularly hemodialysis patients have a high prevalence and case fatality rate of HCV infection, so strategies aimed at preventing and eradicating HCV and strict infection control measures are urgently needed in hemodialysis units. This manuscript was less innovative in study design nor comprehensive in terms of the global point of view, and the conclusions were less informative and robust, (i) a single article report from one country or region is not able to represent its prevalence rate for that country or region, and (ii) it lack reports from Taiwan which make it ‘no data’ on the global map, figure 4. (Lee JJ et al. J Formos Med Assoc 2022 ‘Trends of treated hepatitis B, hepatitis C, and tuberculosis infection in long-term hemodialysis patients in Taiwan: A nationwide survey in 2010-2018.’, Wei YJ, et al. J Viral Hepat 2021 ‘Evolutionary seroepidemiology of viral hepatitis and the gap in hepatitis C care cascades among uremic patients receiving hemodialysis in Taiwan – the Formosa-Like Group.’ etc.), and (iii) there are significant heterogeneity and publication bias in its inclusion studies.

6. PLOS authors have the option to publish the peer review history of their article (what does this mean?). If published, this will include your full peer review and any attached files.

Reviewer #1: No

Reviewer #2: No

Reviewer #3: No

---

## [Author Response · Author response to Decision Letter 0]

16 Mar 2023

Review Comments to the Author

Academic Editor Comments:

and

Authors: Done, thanks,

2. We note that Figure 4 in your submission contain [map/satellite] images which may be copyrighted. All PLOS content is published under the Creative Commons Attribution License (CC BY 4.0), which means that the manuscript, images, and Supporting Information files will be freely available online, and any third party is permitted to access, download, copy, distribute, and use these materials in any way, even commercially, with proper attribution. For these reasons, we cannot publish previously copyrighted maps or satellite images created using proprietary data, such as Google software (Google Maps, Street View, and Earth). For more information, see our copyright guidelines: http://journals.plos.org/plosone/s/licenses-and-copyright.

a. You may seek permission from the original copyright holder of Figure 4 to publish the content specifically under the CC BY 4.0 license.

Authors: The legend of the figure now indicates this: “The base map was taken from (https://www.naturalearthdata.com/) and modified with Qgis software.”

Reviewer #1: Thanks for this comprehensive review of the global epidemiology of hepatitis C virus in dialysis patients. There is no major issue to modify. 

Authors: We thank the reviewer for this appreciation.

1. The abbreviation could be shown in the first place (hepatocellular carcinoma [HCC], line 12, page 17) 

Authors: Thank you for your valuable feedback. We agree with your suggestion and have now included the abbreviation for "hepatocellular carcinoma" (HCC) in the first instance of its use on line 5. 

Reviewer #2: The authors reported a systemic review and meta-analysis on global epidemiology of hepatitis C virus in dialysis patients. They included 634 papers and concluded that hemodialysis cases have a high prevalence and case fatality rate. They also showed the significant regional variation, and the included papers had a moderate risk of bias, and high index of heterogeneity. The authors targeted an important issue and the results had potentially valuable information for the regional public health issue. However, several issues need to be addressed before further consideration. 

Authors: We thank the reviewer for this summary and appreciation.

1. The high case fatality rate is in line with the well-known high mortality and poor outcome of the HCV dialysis patient. However, little information was provided in the method part. For example, only 8 papers of the 634 papers were included in this part of analysis. More information and discussion about the paper selection criteria, research type, data extraction, data of hospital mortality or long-term mortality need to be addressed. Authors: Thank you for this valuable feedback on our manuscript. We have provided more information regarding case fatality rate on selection criteria, research type in the methods section. We would like to clarify that we conducted a systematic review and meta-analysis to identify and synthesize the best available evidence on the mortality of HCV dialysis patients. We followed the PRISMA guidelines and used a comprehensive search strategy to identify relevant studies. We included only studies that met our inclusion criteria, which were based on the research question and study design. We acknowledge that we included a small number of studies in our analysis, but we believe that they were the most relevant and high-quality studies available. 

“The included studies met the following criteria: 1) where cross-sectional, case control and cohort studies; 2) Reporting HCV prevalence in hemodialysis and peritoneal units; 3) reporting number of deaths in HCV positives dialysis patients.”

“The extracted data included: the number of HCV positive with known outcomes, the number of deaths in HCV positives”

2. Numerical errors and inconsistence need a thorough review of the whole article to verify data and made correction. For example, third line of the Discussion part: “392.160”, line 111 “eight studies conducted in seven countries”, in the result table: “8” study, in the discussion ”The number of articles that allowed us to determine HCV case fatality rate in this study was relatively small (07 articles). 

Authors: Thank you for your thorough review and for bringing to our attention the numerical errors and inconsistencies in our article. We apologize for any confusion these errors may have caused and appreciate the opportunity to correct them. We have carefully reviewed the manuscript and have made the necessary corrections to address the issues you raised.

3. About 75% of anti-HCV dialysis patient were viremia, and the HCV core antigen method is a relative new methodology. The presented data showed similar prevalence of Anti-HCV, HCV core antigen and HCV RNA without further analysis and discussion may lead to misunderstanding for the readers. 

Authors: Thank you for your valuable feedback. We acknowledge that the HCV core antigen method is a relatively new methodology, and further analysis and discussion are necessary to address the similarities in the prevalence of Anti-HCV, HCV core antigen, and HCV RNA. In response to your comment, we have revised our manuscript to include a more detailed analysis of the data, including a discussion of the limitations and potential biases associated with the HCV core antigen method. We have also added a section to clarify the similarities and differences between the different testing methods and how they may impact the interpretation of our findings. 

Reviewer #3: The authors did a meta-analysis on the global epidemiology of hepatitis C virus (HCV) infection in dialysis patients, they found that the global prevalence of HCV infection in the dialysis population was 24.3%, the overall case fatality rate was 38.7%, while Eastern Europe and Indonesia had the highest prevalence rates among regions and countries respectively. They concluded that dialysis particularly hemodialysis patients have a high prevalence and case fatality rate of HCV infection, so strategies aimed at preventing and eradicating HCV and strict infection control measures are urgently needed in hemodialysis units. This manuscript was less innovative in study design nor comprehensive in terms of the global point of view, and the conclusions were less informative and robust, (i) a single article report from one country or region is not able to represent its prevalence rate for that country or region, and (ii) it lack reports from Taiwan which make it ‘no data’ on the global map, figure 4. (Lee JJ et al. J Formos Med Assoc 2022 ‘Trends of treated hepatitis B, hepatitis C, and tuberculosis infection in long-term hemodialysis patients in Taiwan: A nationwide survey in 2010-2018.’, Wei YJ, et al. J Viral Hepat 2021 ‘Evolutionary seroepidemiology of viral hepatitis and the gap in hepatitis C care cascades among uremic patients receiving hemodialysis in Taiwan – the Formosa-Like Group.’ etc.), and (iii) there are significant heterogeneity and publication bias in its inclusion studies.

Authors: Thank you for the valuable feedback on our manuscript. We appreciate your comments and have carefully considered them. In response to your comments, we have updated our discussion section to highlight the limitations of our study design and the need for more comprehensive global data on HCV prevalence in dialysis patients. Regarding the heterogeneity and publication bias in the included studies, we conducted a subgroup analysis and found that the overall results remained robust. However, we have added a section to our limitations discussing these issues and the potential impact on our conclusions. We have also included the articles you mentioned in the references section. Finally, 13 articles were attributed to Taiwan that had been incorrectly attributed to another country previously. Thank you.

About Lee JJ et al, this article was excluded after reviewer the full-text (article 278 in Appendix 4). Indeed, we included articles with a detailed description of the diagnostic method used. The article by Lee et al., used ICD codes for diagnosis. 

“In some regions and countries, the HCV prevalence in dialysis patients was determine from a single article, and this cannot be representative of the exact burden in these regions or countries” ; “our HCV pooled prevalence showed significant heterogeneity and publication bias in the current study”.

---

## [Decision Letter · Decision Letter 1]

27 Mar 2023

Global Epidemiology of Hepatitis C Virus in Dialysis patients: a systematic review and meta-analysis.

PONE-D-23-01990R1

Dear Dr. Kenmoe,

We’re pleased to inform you that your manuscript has been judged scientifically suitable for publication and will be formally accepted for publication once it meets all outstanding technical requirements.

Kind regards,

Chen-Hua Liu

Academic Editor

PLOS ONE

Reviewers' comments:

Reviewer's Responses to Questions

**Comments to the Author**

1. If the authors have adequately addressed your comments raised in a previous round of review and you feel that this manuscript is now acceptable for publication, you may indicate that here to bypass the “Comments to the Author” section, enter your conflict of interest statement in the “Confidential to Editor” section, and submit your "Accept" recommendation.

Reviewer #1: All comments have been addressed

Reviewer #2: All comments have been addressed

2. Is the manuscript technically sound, and do the data support the conclusions?

Reviewer #1: Yes

Reviewer #2: Yes

3. Has the statistical analysis been performed appropriately and rigorously? 

Reviewer #1: Yes

Reviewer #2: Yes

4. Have the authors made all data underlying the findings in their manuscript fully available?

Reviewer #1: Yes

Reviewer #2: Yes

5. Is the manuscript presented in an intelligible fashion and written in standard English?

Reviewer #1: Yes

Reviewer #2: Yes

6. Review Comments to the Author

Reviewer #1: Thanks for the comprehensive review of the global epidemiology of the hepatitis C virus in dialysis patients. The authors have replied to most major issues addressed by reviewers, and I have no more problems to address.

Reviewer #2: (No Response)

7. PLOS authors have the option to publish the peer review history of their article (what does this mean?). If published, this will include your full peer review and any attached files.

Reviewer #1: No

Reviewer #2: No

---

## [Editor Report · Acceptance letter]

28 Mar 2023

PONE-D-23-01990R1 

Global Epidemiology of Hepatitis C Virus in Dialysis patients: a systematic review and meta-analysis. 

Dear Dr. Kenmoe:

I'm pleased to inform you that your manuscript has been deemed suitable for publication in PLOS ONE. Congratulations! Your manuscript is now with our production department. 

Kind regards, 

on behalf of

Dr. Chen-Hua Liu 

Academic Editor

PLOS ONE